# Chemical- and photo-activation of protein-protein thiol-ene coupling for protein profiling
André Campaniço[1], Marcin Baran[2], Andrew G. Bowie[2], Daniel B. Longley[3], Timothy Harrison[3] & Joanna F. McGouran [1]✉

The thiol-ene reaction between an alkene and a thiol can be exploited for selective labelling of cysteine residues in protein profiling applications. Here, we explore thiol-ene activation in systems from chemical models to complex cellular milieus, using UV, visible wavelength and redox initiators. Initial studies in chemical models required an oxygen-free environment for efficient coupling and showed very poor activation when using a redox initiator. When thiol-ene activation was performed in protein and cell lysate models, all three initiation methods were successful. Faster thiol-ene reaction was observed as the cysteine and alkene were brought into proximity by a binding event prior to activation, leading to quicker adduct formation in the protein model system than the chemical models. Furthermore, in the protein-protein coupling, none of the activators required an oxygen-free environment. Taken together, these observations demonstrate the broad potential for thiol-ene coupling to be used in protein profiling.

The thiol-ene coupling was first described in 1905[1] and, since then, it has been used in diverse applications including polymer synthesis[2–4], peptide chemistry[5–8], synthetic chemistry[9–11] and, more recently, protein labelling[12]. It comprises the efficient reaction between a thiol and a carbon-carbon double bond, leading to the formation of a thioether[13]. Thiol radical activation is generally accomplished through photochemical activation[13,14]. Several photo-initiators have been reported to activate this system, including UV, visible and infrared radiation to catalyse the coupling[2,15,16]. Redox activation of thiols has been scarcely reported, although transition metals are known to activate hydroxyl radicals[17,18]. The thiol-ene coupling is considered a 'click' reaction due to its typical high selectivity and efficiency, and lack of side-products[13,19]. These characteristics make it suitable for the labelling of cysteine residues, as it has been previously reported in the labelling of deubiquitinating enzymes (DUBs) using ubiquitin-based activity-based probes (ABPs)[12].

Ubiquitination is a key posttranslational protein modification within the cell. Ubiquitin, a 76 amino acid protein, is added to substrate proteins by the E1/E2/E3 ligase-system, modulating their function, location and degradation[20–22]. The removal of ubiquitin from substrates is carried out by DUBs, a family of proteins with about 100 members, divided into seven subfamilies, six of which are cysteine proteases[23–26]. These catalyse the hydrolysis of the isopeptide bond between the C-terminus of ubiquitin and the protein substrate using their active-site thiol (Fig. 1a)[27,28]. Disruption of DUB activity has been linked to disease states including cancer[29–32], neurodegeneration[33–35] and inflammation[36,37]. Ubiquitin-based ABPs are a valuable tool in the study of DUBs, taking advantage of the DUB active-site cysteine to label this class of proteins[38,39]. More recently, an alkene-based ubiquitin-ABP (HA-[1-75]Ub-alkene probe **1**, Fig. 1b) was developed, exploiting thiol-ene chemistry to controllably activate this labelling. An alkene moiety was installed in the place of the scissile C-terminal isopeptide bond, aligning the warhead to react with the DUB active-site thiol in the $C_2$ position (Fig. 1b)[12,40–42]. Initial studies used a mixture of 2,2-dimethoxy-2-phenylacetophenone (DPAP) as the radical initiator and 4'-methoyacetophenone (MAP) as the photosensitiser. Degassing was observed to be a critical step and the coupling occurred upon UV exposure[12]. Efforts were made to carry out the same reaction with white light, using Eosin Y, however, even though the coupling was efficient using recombinants DUBs, no significant labelling was observed in complex cell lysate mixtures[43].

In this work, a broad study of thiol-ene coupling and its different activation methods for protein profiling was performed. HA-[1-75]Ub-alkene probe **1** and its interaction with the DUB active site were used as protein models. Probe **1** is a validated tool for the study of DUBs and its unique thiol-ene-triggered activation acts as the ideal model for the study of this system between bound proteins[12,41]. Thiol-ene activation was catalysed using UV light, visible light and redox activation. 2-Hydroxy-4'-(2-hydroxyethoxy)-2-methylpropiophenone (Irgacure 2959)[44,45] and 9-mesityl-10-

[1]School of Chemistry, Trinity College Dublin, Dublin, Ireland. [2]School of Biochemistry and Immunology, Trinity College Dublin, Dublin, Ireland. [3]The Patrick G. Johnston Centre for Cancer Research, Queen's University Belfast, Belfast, UK. ✉e-mail: jmcgoura@tcd.ie

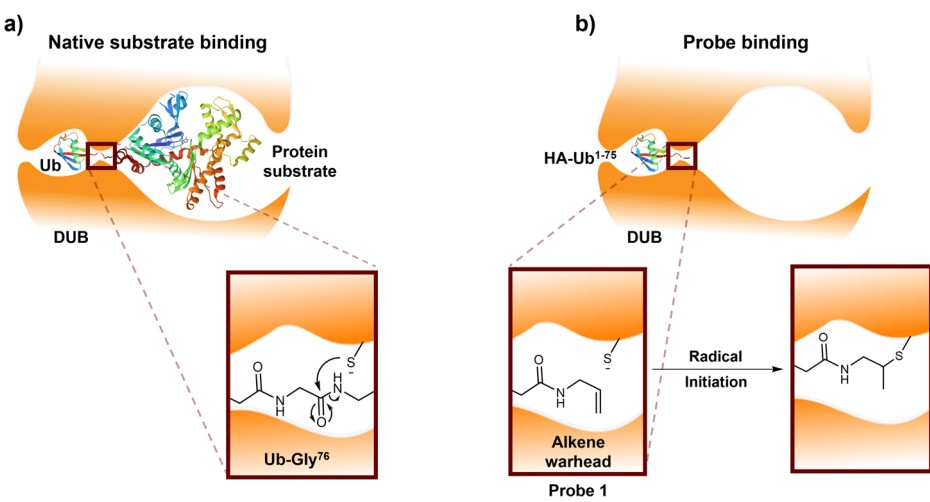

**Fig. 1 | DUB active-site with wild-type ubiquitin coupled to a protein substrate and with HA-[1-75]Ub-alkene probe 1. a** Cleavage of the isopeptide bond between Gly[76] from ubiquitin (Ub) and a protein substrate, by the DUB active-site cysteine. **b** Proposed interaction between the HA-[1-75]Ub-alkene probe **1** and the DUB active site. The Ub-Gly[76] residue was replaced with an inert alkene warhead to generate a ubiquitin-based activity-based probe. Upon radical activation of the DUB active-site thiol, HA-[1-75]Ub-alkene probe **1** undergoes thiol-ene coupling, forming a new C-S bond between the probe and the DUB.

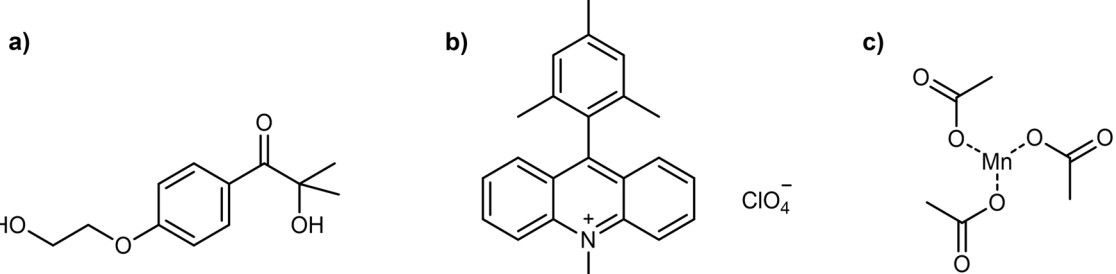

**Fig. 2 | Initiators used to trigger the thiol-ene coupling in this study. a** Irgacure 2959; **b** 9-mesityl-10-methylacridinium perchlorate (Mes-Acr[+]); **c** manganese (III) acetate.

methylacridinium perchlorate (Mes-Acr[+])[16,46] were selected as the light-induced radical initiators (Fig. 2a, b, Fig. S1a, b), due to their proven ability to catalyse the thiol-ene coupling in peptides. Irgacure 2959 and Mes-Acr[+] are triggered by UV and blue wavelengths, respectively. In an attempt to induce the thiol-ene coupling through a redox mechanism, manganese acetate was selected as the redox initiator (Fig. 2c, Fig. S1c), due to its previously demonstrated ability to activate thiols[47]. An initial study was performed with the three initiators using chemical models. The light-induced initiators led to more efficient conversions, with Irgacure 2959 affording the highest yield. The reaction with Mes-Acr[+] was much slower, with significant recovery of starting material after the given reaction time but high calculated conversion into the desired product. Redox activation led to disulfide formation, with little thiol-ene conversion. In the chemical model, degassing significantly improved product formation for all initiators. The initiators were then tested with probe **1** in complex cell lysate mixtures and the labelling conditions were optimised. All initiators were successful in triggering the thiol-ene coupling using these models. The role of degassing was explored in the protein models. Contrary to what was observed in the chemical models, only the UV initiator showed improved labelling with this additional step. Both Mes-Acr[+] and Mn(OAc)₃ had decreased thiol-ene efficiencies in oxygen-free media. For Irgacure 2959, although degassing improves the labelling signal, it is not essential for the reaction to occur as labelling can still be observed without it. UV light activation was the most efficient, with the thiol-ene occurring with only 2 min of irradiation. Satisfyingly, activation with blue wavelengths can be performed in the same timeframe by increasing LED intensity. Redox activation afforded significant labelling after 5 min of incubation with the initiator. However, maximum labelling was only observed after 1 h and, therefore, redox activation was deemed less efficient than light-mediated activation. Following protein-protein thiol-ene optimisation, probe **1** was then incubated with different recombinant DUB

models and with the non-DUB proteins bovine serum albumine (BSA) and β-galactosidase to demonstrate the selectivity and pre-binding requirement for labelling under these conditions. For all initiators, thiol-ene was successful with the selected recombinant DUBs, but not with BSA and β-galactosidase, supporting the selectivity of this system.

## Results and discussion
### Preliminary studies with chemical models
A preliminary study using the selected UV, blue light and redox initiators (Irgacure 2959, Mes-Acr[+] and Mn(OAc)₃) was performed with simple thiol-ene substrates. N-Boc-L-cysteine methyl ester (**2**) and allyl alcohol (**3**) were chosen as the thiol-ene model system. Negative and positive controls were performed using no initiator (Entries 1–2, Table 1) and with a mixture of DPAP and MAP under standard conditions[12] (Entry 3, Table 1). The thiol-ene product **4** was obtained with DPAP and MAP, with an isolated yield of 64%, after 30 min of UV irradiation. The reaction was performed with Irgacure 2959, with 30-min UV irradiation. The thiol-ene product was obtained, and it was observed that degassing of the mixture prior to UV irradiation led to a yield increase from 18% to 84% (Entry 4 vs 5, Table 1), displaying a higher efficiency than the positive control DPAP/MAP. Next, the reaction was carried out with Mes-Acr[+] and blue LED irradiation. The first attempt was performed in ethanol, but Mes-Acr[+] was poorly soluble in this solvent and no reaction was observed (Entry 6, Table 1). The solvent was altered to acetonitrile, but still, without the degassing step, no reaction was observed (Entry 7, Table 1). After degassing the mixture, the product was obtained with a low yield of 3% along with 95% starting material recovery (Entry 8, Table 1). Although the isolated yield was low, 95% of starting material 2 was recovered. From the 5% starting material consumed in the reaction, the isolated yield of 3% of 4 represents a 69% conversion of the consumed starting material into desired product. The reaction was next

**Table 1 | Study of selected initiators with thiol-ene chemical model system**

| Entry | Initiator | Light source | Solvent | Degassing | Reaction time | Yield 4 | Recovered 2 | Conversion 2 into 4 | Disulfide Yield |
|---|---|---|---|---|---|---|---|---|---|
| 1 | - | - | EtOH | No | 30 min | NR | - | - | - |
| 2 | - | - | EtOH | Yes | 30 min | NR | - | - | - |
| 3 | DPAP/MAP | UV | EtOH | Yes | 30 min | 64% | - | 64% | - |
| 4 | Irgacure 2959 | UV | EtOH | No | 30 min | 18% | - | 18% | - |
| 5 | Irgacure 2959 | UV | EtOH | Yes | 30 min | 84% | - | 84% | - |
| 6 | Mes-Acr$^+$ | Blue Light | EtOH | No | 30 min | NR | - | - | - |
| 7 | Mes-Acr$^+$ | Blue Light | ACN | No | 30 min | NR | - | - | - |
| 8 | Mes-Acr$^+$ | Blue Light | ACN | Yes | 30 min | 3% | 95% | 69% | - |
| 9 | Mes-Acr$^+$ | Blue Light | ACN | Yes | 6 hours | 12% | 85% | 80% | - |
| 10 | Mn(OAc)$_3$ | - | EtOH | No | 30 min | NR[a] | - | - | 58% |
| 11 | Mn(OAc)$_3$ | - | EtOH | Yes | 30 min | 3%[a] | - | 3% | 46% |

N-Boc-L-cysteine methyl ester (2) and allyl alcohol (3) were reacted under thiol-ene coupling conditions.
*NR* no reaction.
[a]disulfide formation was observed.

attempted for 6 h instead of the initial 30 min, with a yield improvement to 12% and starting material recovery of 85%, affording a conversion of **2** into **4** of 80% (Entry 9, Table 1). Although Mes-Acr$^+$ was not as efficient as the UV initiators tested (DPAP/MAP and Irgacure 2959) and required extended reaction times, the use of Mes-Acr$^+$ did afford high conversions into product **4** and low side-product formation. Finally, the coupling was attempted with Mn(OAc)$_3$, as a redox initiator, with an incubation time of 30 min. When the reaction was carried out without a degassing step to remove atmospheric oxygen, only disulfide formation was observed in a yield of 58% (Entry 10, Table 1). A second attempt was performed with a degassing step. The thiol-ene product **4** was obtained in a low yield of 3% and a disulfide yield of 46% was still observed (Entry 11, Table 1). Contrary to the light-dependent initiators, the redox initiator favoured disulfide formation over the thiol-ene product, however small amounts of the competing thiol-ene product could be observed in degassed samples. Interestingly, for the initiators tested, degassing was observed to be a critical step for the thiol-ene coupling, as it led to a significant yield increase for Irgacure 2959 and to thiol-ene product formation for Mes-Acr$^+$ and Mn(OAc)$_3$. Following this study, Irgacure 2959 and Mes-Acr$^+$ were considered suitable for more complex biological models due to the high yield and good conversions observed in the model systems. The Mn(OAc)$_3$ study indicated that it could be used to catalyse the thiol-ene reaction provided disulfide formation could be suppressed. It was reasoned that a protein-protein thiol-ene reaction, in which there was a binding event prior to thiol-ene initiation, would favour the thiol-ene reaction over disulfide formation due to the induced proximity of the thiol and alkene. To test this hypothesis, we included an additional model system to investigate a protein-protein thiol-ene reaction using model enzyme OTUB1. In this system, we anticipated that binding interactions prior to radical initiation could be exploited to favour the thiol-ene product over disulfide formation.

**Additional preliminary studies with Mn(OAc)$_3$ and OTUB1**

Although Mn(OAc)$_3$ displayed limited efficiency in triggering the thiol-ene coupling in the chemical model system (Entry 11, Table 1), Mn(OAc)$_3$ has been shown to effectively promote thiol activation[47], leading to disulfide formation in the model system. To assess if induced proximity between the thiol and the alkene may contribute to suppression of disulfide formation and efficient thiol-ene coupling, a second model system was employed with the HA-$^{1-75}$Ub-alkene probe **1** and the recombinant DUB OTUB1 (OTU Domain-containing ubiquitin aldehyde-binding protein 1), a system previously validated using DUBs with the thiol-ene coupling using DPAP and MAP[12]. Probe **1** was incubated with recombinant OTUB1 and different concentrations of Mn(OAc)$_3$ (Fig. 3). A negative control with no initiator and a positive control with 500 µM of DPAP/MAP were used (Lanes 3 and 4, Fig. 3). Only the positive control was degassed. Significant labelling of OTUB1 with the HA-$^{1-75}$Ub-alkene probe **1** was observed at all concentrations tested between 125 µM and 2 mM (Lanes 5-9, Fig. 3). Similar labelling was observed in all lanes, suggesting that 125 µM of this initiator is sufficient for an adequate thiol-ene triggering between these two protein models. A second assay was performed with lower Mn(OAc)$_3$ concentrations (Supplementary Fig. S2), however none of the concentrations tested below 125 µM showed significant labelling. The discrepancy between this assay, in which Mn(OAc)$_3$ could effectively trigger thiol-ene coupling, and the chemical models, in which very little thiol-ene product was observed, was attributed to the interactions between the ubiquitin-based probe **1** and recombinant OTUB1. Prior to Mn(OAc)$_3$ addition, probe **1** was incubated with OTUB1 to allow binding of the ubiquitin motif within the deubiquitinating enzyme, aligning the alkene warhead in the optimal position for the thiol-ene to occur upon thiol activation. The proximity between the thiol and alkene following protein-protein binding leads to an efficient coupling, demonstrating that the occurrence of thiol-ene in systems with induced thiol-alkene proximity display different characteristics and may be prone to activation methods that are less effective in chemical batch models. Following this result, Mn(OAc)$_3$ was considered suitable for

further studies and used in more complex biological mixtures with Irga-cure 2959 and Mes-Acr$^+$.

## Optimisation of the thiol-ene system for protein profiling

As all initiators were successful either with the chemical models or the OTUB1 model, to investigate their potential to trigger thiol-ene coupling in complex biological systems, HA-$^{1-75}$Ub-alkene probe **1** was incubated with HEK293T cell lysates, using Irgacure 2959, Mes-Acr$^+$ and Mn(OAc)$_3$. In this assay, probe **1** is anticipated to bind to endogenous DUBs, placing the alkene moiety in probe **1** in proximity to the active site cysteine of the bound

DUB. Radical activation of the active site thiol would then result in covalent capture of any bound DUBs. Apart from initiator efficiency, this assay was used to investigate the biocompatibility of each of the initiators within complex cellular mixtures.

## Role of light-irradiation and degassing

To demonstrate the requirement for light-irradiation in the activation of Irgacure 2959 and Mes-Acr$^+$, the thiol-ene coupling was attempted with these initiators, using probe **1** and HEK293T cell lysates, with and without light-irradiation. When the reaction mixtures were kept in the dark, no labelling was observed (Lanes 5 and 7, Fig. 4a) while light-irradiated samples afforded efficient thiol-ene labelling (Lanes 4 and 6, Fig. 4a). This result confirms that the radical activation promoted by Irgacure 2959 and Mes-Acr$^+$ is triggered by light, allowing for spatial and temporal control of the activation step.

The role of degassing was then investigated for the three initiators. Degassing had a significant effect in the chemical models and was essential for the triggering of thiol-ene using DPAP/MAP in protein-protein inter-actions in previous studies[12]. However, when possible, degassing-dependent initiators should be avoided as it can cause protein denaturation and reduce biocompatibility. To investigate the need for degassing, probe **1** was incubated with HEK293T cell lysates prior to initiation. For each initiator, a control with a non-degassed sample was used and compared with a degassed sample (Fig. 4b and Supplementary Fig. S3). In the samples with Irgacure 2959, it was observed that the degassing step led to stronger protein labelling (Lanes 4–5, Fig. 4b). It was however observed that, unlike in previous studies with DPAP/MAP, it was not essential for the thiol-ene coupling, as sig-nificant thiol-ene labelling was still observed in the non-degassed sample. This result suggests that the need for an oxygen-free environment might be initiator-dependent. For Mes-Acr$^+$ and Mn(OAc)$_3$, contrary to what was observed in the chemical models, sample degassing led to weaker labelling (Lanes 6–9, Fig. 4b). This is not the first report of thiol-ene occurring in the presence of oxygen as other reports have proposed conditions for thiol-ene and thiol-yne polymerisation that are insensitive to the presence of oxygen[3,48,49] or that benefit from it[50]. More specifically, previous studies of

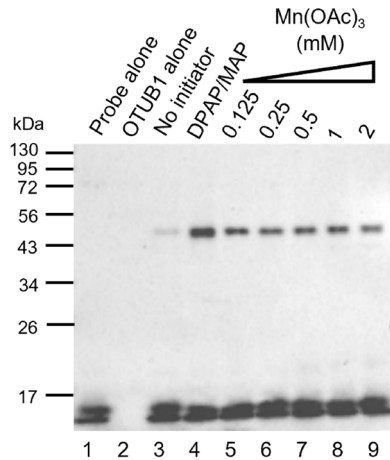

**Fig. 3 | Anti-HA Western Blot analysis of the incubation of the HA-$^{1-75}$Ub-alkene probe 1 with the recombinant DUB OTUB1.** HA-$^{1-75}$Ub-alkene probe **1** (2 ug) was incubated with the DUB OTUB1 (1 ug) for 90 min at 37 ºC, prior to addition of Mn(OAc)$_3$ (125 μM - 2 mM) and a further 30 min incubation. A negative control without Mn(OAc)$_3$ and a positive control using a mixture of DPAP and MAP at a concentration of 500 μM were performed. The positive control was degassed and irradiated with UV, for 2 min. The results were analysed by SDS-PAGE and visualised by anti-HA Western Blot.

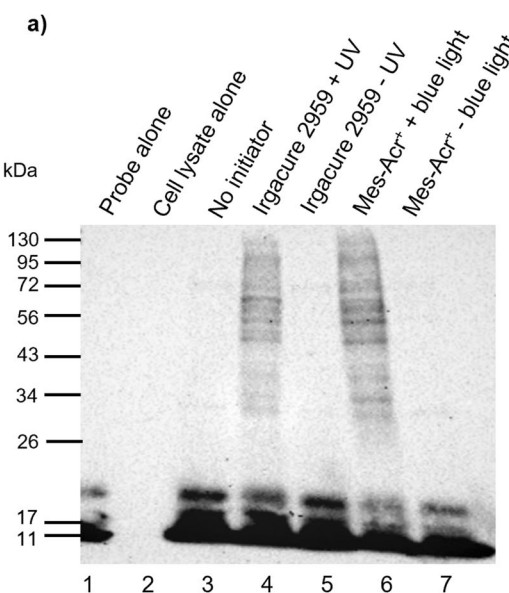

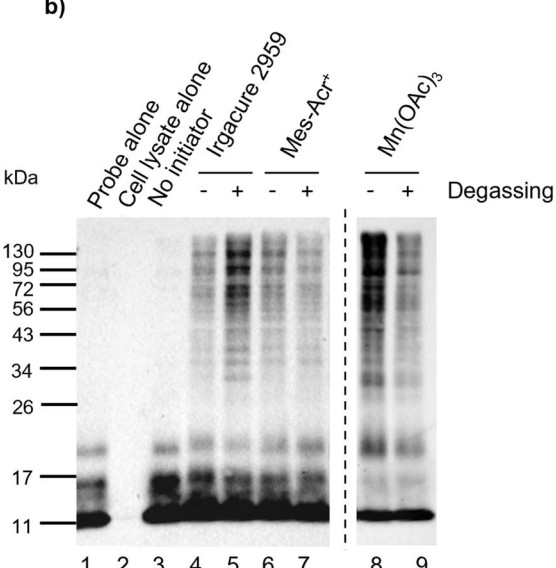

**Fig. 4 | Anti-HA Western Blot analysis of the labelling of HEK293T cell lysates with HA-$^{1-75}$Ub-alkene probe 1, under non-irradiating and degassing conditions.** HA-$^{1-75}$Ub-alkene probe **1** (2 μg) was incubated with HEK293T cell lysates (50 μg), for 90 minutes at 37 ºC, prior to initiator addition. A negative control without initiator was performed. Results were analysed by SDS-PAGE and visualised by anti-HA Western Blot. **a** Initiators Irgacure 2959 (250 μM, 2-min UV irradiation) and

Mes-Acr$^+$ (100 μM, 10-min blue light irradiation) were tested under light-irradiation and non-irradiation conditions, **b** Role of degassing for initiators Irgacure 2959 (250 μM, 2-min UV irradiation), Mes-Acr$^+$ (100 μM, 10-min blue light irradiation) and Mn(OAc)$_3$ (5 mM, 30-min incubation at 37 ºC) was investi-gated. Non-degassed samples were used as the control and were compared with degassed samples.

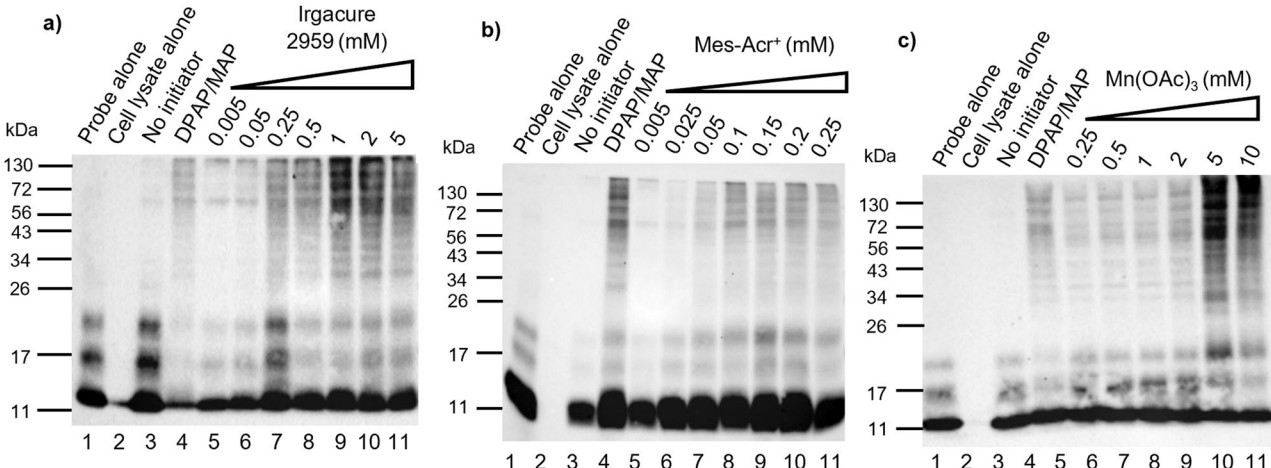

**Fig. 5 | Anti-HA Western Blot analysis of the incubation of the HA-$^{1-75}$Ub-alkene probe 1 with HEK293T cell lysates, using different concentrations of initiators.** HA-$^{1-75}$Ub-alkene probe **1** (2 µg) was incubated with HEK293T cell lysates (50 µg) for 90 min at 37 °C, prior to initiator addition. A negative control without initiator and a positive control using a mixture of DPAP and MAP at a concentration of 500 µM were performed. The positive control was degassed and irradiated with UV, for 2 min. Results were analysed by SDS-PAGE and visualised by anti-HA Western Blot. **a** A range of concentrations of Irgacure 2959 from 5 µM to 5 mM were tested. Following Irgacure 2959 addition, the samples were irradiated with UV for 2 min, without previous degassing, **b** A range of concentrations of Mes-Acr$^+$ from 5 µM to 250 µM were tested. The samples were irradiated with a blue LED for 5 min, without previous degassing, **c** A range of concentrations of Mn(OAc)$_3$ from 250 µM to 10 mM were tested. Following Mn(OAc)$_3$ addition, the samples were incubated at 37 °C for 30 min. without sample degassing.

protein-protein thiol-ene between probe **1** and the DUB OTUB1 using Eosin Y and white light showed a similar result, with degassing decreasing the labelling intensity[43]. As none of the initiators tested required degassing for labelling to occur, all subsequent assays omitted this step, with the exception of DPAP/MAP initiated samples, simplifying the procedure and increasing biocompatibility.

### Initiator concentration dependence

To test the effect of initiator concentration, varying concentrations of Irgacure 2959, Mes-Acr$^+$ and Mn(OAc)$_3$ were added following incubation between probe **1** and the lysate. A negative control with no initiator and a positive control with 500 µM of DPAP/MAP were used (Lanes 3–4, Fig. 5a–c). As the thiol-ene protein labelling using Irgacure 2959, Mes-Acr$^+$ and Mn(OAc)$_3$ proved to be relatively insensitive to atmospheric oxygen, only the positive control was degassed. Pleasingly, all three initiators showed significant protein labelling, comparable to the DPAP/MAP positive control (Fig. 5a–c). The samples with Irgacure 2959 were irradiated with UV for 2 min, following the irradiation time previously optimised for DPAP/MAP[12]. Although maximum labelling was obtained at the concentration of 1 mM (Lane 9, Fig. 5a), significant lysate labelling can be observed using only 250 µM (Lane 7, Fig. 5a) and that was selected as the optimal concentration, in an attempt to limit excess radical formation. Radical formation may lead to changes in the protein backbone, inducing protein denaturation and degradation[51]. These results demonstrated that Irgacure 2959 successfully initiated thiol-ene labelling of HEK293T lysates with a 2-fold decrease in the initiator concentration required in comparison to DPAP/MAP. Furthermore, this was achieved without the requirement for degassing. Samples with Mes-Acr$^+$ were irradiated with a blue LED for 5 min. In the first assay performed, only mild labelling was observed at 50 µM, with protein degradation being observed at concentrations of 500 µM and higher (Supplementary Fig. S4).

To more closely investigate initiator concentration, a new assay was performed with a concentration range of 5–250 µM (Lanes 5-11, Fig. 5b). Significant labelling was observed between 100 and 200 µM, with no significant differences observed among these concentrations (Lanes 8-10, Fig. 5b). Mes-Acr$^+$ compared favourably to DPAP/MAP in this assay, showing significant labelling with a 5-fold decrease in initiator concentration, without the need for an oxygen-free environment (Lane 8 vs 4, Fig. 5b). Finally, samples with the redox initiator Mn(OAc)$_3$ were incubated at 37 °C, for 30 minutes. Mild labelling was observed at the concentration of 2 mM and below (Lanes 5-9, Fig. 5c), with significant labelling observed only at

concentrations of 5 mM and 10 mM (Lanes 10-11, Fig. 5c). Mn(OAc)$_3$ was the least efficient initiator, requiring higher concentrations for thiol-ene labelling to occur. Despite requiring a minimum concentration of 5 mM in this model, redox thiol-ene activation was successful in this protein model. Discrepancies between the incubation of probe **1** with OTUB1 and HEK293T cell lysates were observed, as 125 µM of Mn(OAc)$_3$ was enough to label OTUB1 and a 40-fold concentration increase was required for the lysates. Manganese (III) is known to interact with carbonyl moieties, inducing α-carbon deprotonation and enol formation, and consequent radical formation at the α-carbon[52–55]. Its reactivity has been demonstrated with ketones, carboxylic acids, esters and amides, suggesting that manganese (III) may interact with several biological macromolecules, including proteins. Despite its potential toxicity, protein degradation was not observed in any of the concentrations tested. However, this potential manganese side reactivity in complex cellular mixtures, may limit the use of Mn(OAc)$_3$ as an initiator in a complex cellular milieu.

### Initiator incubation time

To further optimise the thiol-ene coupling within cell lysates using the selected initiators, different irradiation times for Irgacure 2959 and Mes-Acr$^+$ and different incubation times for Mn(OAc)$_3$ were analysed. The optimal initiator concentrations of 250 µM for Irgacure 2959, 100 µM for Mes-Acr$^+$ and 5 mM for Mn(OAc)$_3$ were used. In each instance, labelling was compared with a negative control with no initiator and a positive control with a mixture of 500 µM of DPAP/MAP (Lane 3–4, Fig. 6a–d). For the samples with Irgacure 2959, irradiation times were selected based on previous studies with DPAP/MAP[12] and optimal labelling was observed after 2 min of UV irradiation (Lane 7, Fig. 6a). UV irradiation is known to promote irreversible protein denaturation, due to formation of free radicals and radical oxygen species leading to changes in primary, secondary and tertiary structures[56,57]. The more susceptible residues to UV light excitation and free radical formation are tryptophan, tyrosine and phenylalanine, due to their aromaticity[56,58]. Though there is no significant difference in DUB labelling between irradiating for 2 min or 5 min, a 2-min irradiation was selected as optimal to shorten UV irradiation time and prevent protein denaturation and degradation. For Mes-Acr$^+$, an initial assay was performed using a blue LED of 36 W and an optimal irradiation time of 10 min was observed (Lane 9, Fig. 6b). To investigate if it was possible to reduce the irradiation time by increasing the LED intensity, a second assay was performed with a blue LED of 90 W. In this setting, an optimal irradiation time

**Fig. 6 | Anti-HA Western Blot analysis of the incubation of the HA-$^{1-75}$Ub-alkene probe 1 with HEK293T cell lysates, using different irradiation/incubation times with the selected initiators.** HA-$^{1-75}$Ub-alkene probe **1** (2 μg) was incubated with HEK293T cell lysates (50 μg) for 90 min at 37 °C, prior to initiator addition. A negative control without initiator and a positive control using a mixture of DPAP and MAP at a concentration of 500 μM were performed. The positive control was degassed and irradiated with UV, for 2 min. Results were analysed by SDS-PAGE and visualised by anti-HA Western Blot. **a** 250 μM of Irgacure 2959 and UV irradiation times between 0.5 and 5 min were tested, **b** 100 μM of Mes-Acr$^+$ and a blue LED of 36 W were used, and irradiation times between 0.5 and 15 min were tested, **c** 100 μM of Mes-Acr$^+$ and a blue LED of 90 W were used, and irradiation times between 0.5 and 15 min were tested, **d** 5 mM of Mn(OAc)$_3$ were used and the mixture was incubated at 37 °C, with incubation times ranging between 1 min and 90 min.

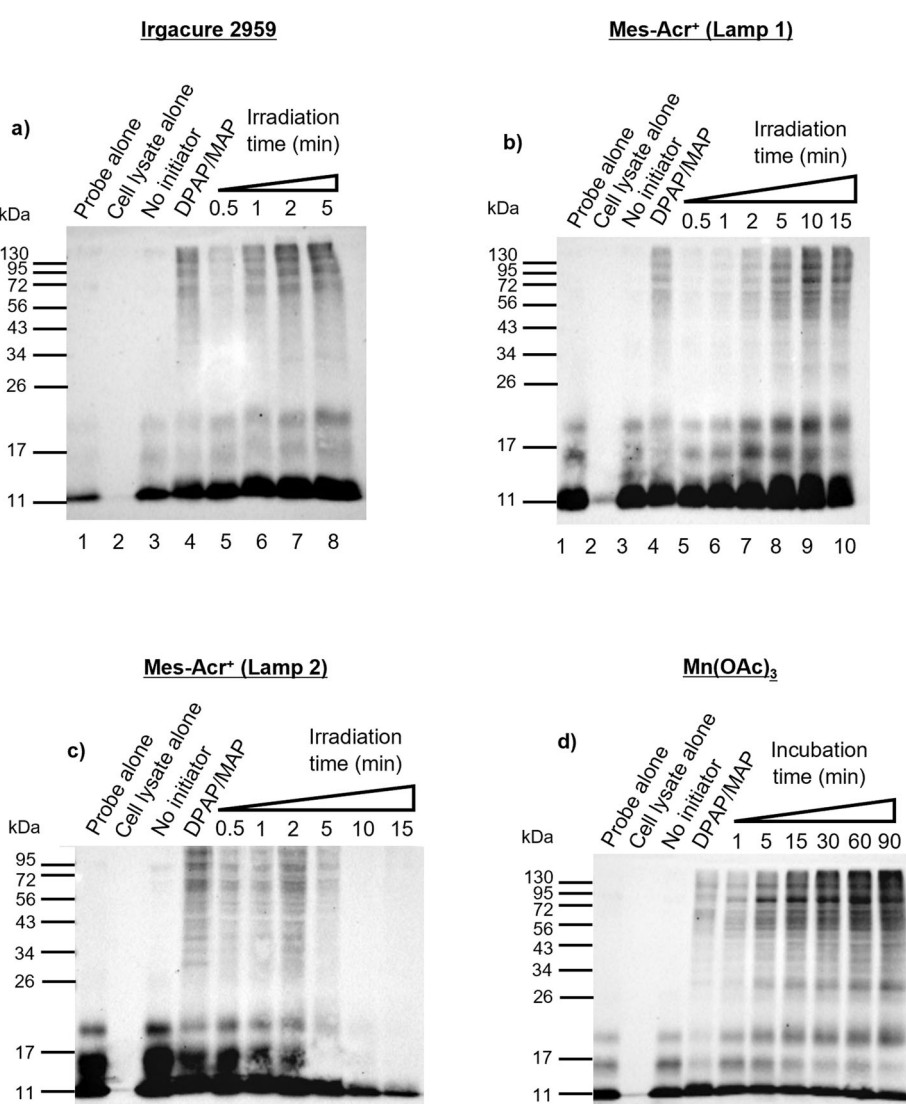

of 2 min was observed (Lane 7, Fig. 6c), however, protein degradation was observed after 5 min (Lane 8-10, Fig. 6c). Blue light irradiation promotes oxidation of proteins through generation of superoxide, singlet oxygen species and hydrogen peroxide. The residues more susceptible to oxidation are cysteine, tyrosine, methionine, histidine and tryptophan[59]. These modifications may have contributed to protein denaturation and degradation at longer irradiation times. Still, these results demonstrate that quick, efficient thiol-ene labelling can be obtained with Mes-Acr$^+$ and Irgacure 2959 in complex biological samples. The short incubation times are in stark contrast to the times required using Mes-Acr$^+$ and Irgacure 2959 in the chemical models. These highly efficient reactions observed in the protein model are rationalised by the binding interactions of probe **1** with endogenous deubiquitinating enzymes in the cell lysate which serve to present the alkene in close proximity to the active site thiol, leading to a fast capture step once the active site thiol is activated. For Mn(OAc)$_3$, longer incubation times of 1–90 min were investigated as it was hypothesised that redox activation might be slower than light-induced radical initiation. Incubation was performed at 37 °C. The optimal incubation time was determined to be 60 min (Lane 9, Fig. 6d). It was however observed that after 5 min of incubation, the labelling was stronger than the labelling observed for the positive control (Lane 6, Fig. 6d). Therefore, despite the optimal incubation time of 1 h, the labelling assay can be performed with this redox initiator using lower incubation times.

## Incubation of probe 1 and recombinant DUB and non-DUB models

A set of final assays were performed using the optimised conditions for each initiator with three recombinant DUBs (OTUB2, USP2 and USP7) and the non-DUBs BSA and β-galactosidase. This assay was conducted to demonstrate the ability of the initiators to promote a protein-protein thiol-ene reaction with a range of deubiquitinating enzymes and demonstrate that the thiol-ene probe-protein adduct formation is binding dependant. BSA was chosen as the non-DUB negative control for this assay, as it displays one free thiol, that could undergo thiol-ene with HA-$^{1-75}$Ub-alkene probe **1** if non-specific labelling was occurring. The selected initiators (Irgacure 2959, Mes-Acr$^+$ and Mn(OAc)$_3$) were added to the probe-protein mixtures, without degassing. Exceptionally, Mn(OAc)$_3$ was added at a concentration of 125 μM, the lower concentration at which labelling was observed with the recombinant DUB OTUB1 (Fig. 3), to investigate whether higher concentrations were only required in complex protein mixtures. Low levels of residual labelling were observed in the negative control with no initiator (lanes 5–7, Fig. 7). However, all initiators were successful in triggering the thiol-ene coupling between probe **1** and the recombinant DUBs selected as significantly stronger labelling bands were seen under these conditions (lanes 8–16 vs lanes 5–7, Fig. 7). Furthermore, stronger labelling was observed with Mn(OAc)$_3$ when compared to Irgacure 2959 and Mes-Acr$^+$, suggesting that redox activation might be more efficient in the labelling of

purified recombinant proteins than light-dependent initiation. Gratifyingly, labelling was not observed using BSA (lanes 9–11, Fig. 8a), even though a larger amount of the non-DUB protein was used in the assay (2.5 μg of BSA vs 1 μg of OTUB2). The same result was obtained with β-galactosidase, with labelling observed only with the DUB used (lanes 5–7 vs lanes 9–11, Fig. S5). The inability of probe **1** to label BSA and β-galactosidase supports probe selectivity for DUBs. A difference in band intensity was observed for BSA between the sample with no initiator (lane 8, Fig. 8b) and the samples with the selected initiators (lanes 9–11, Fig. 8b). This difference was attributed to protein degradation caused by the different conditions used to trigger the thiol-ene coupling.

## Conclusion

In this work, a study of thiol-ene coupling and its different activation methods for protein profiling was performed. Thiol-ene coupling was investigated in chemical models, recombinant proteins and complex cellular mixtures, with significant differences observed. The induced proximity between the alkene moiety and the DUB active-site cysteine thiol achieved in the protein models leads to faster thiol-ene adduct formation than the chemical models, allowing for the use of activation methods which have very limited efficiency in the chemical models. While UV-induced thiol-ene coupling displayed similar efficiency in both models, the visible light- and redox-activated initiators displayed strong thiol-ene labelling in all protein models used, despite their poor results in non-protein-based models. Furthermore, the alignment between the alkene and thiol induced by protein-probe binding prior to activation overcomes the need for an oxygen-free environment, a disadvantageous and limiting step in biological applications. The requirement for such a pre-interaction, which will align the alkene and thiol moieties for efficient thiol-ene reaction, also increases the specificity of the probe, limiting any side reactivity with other thiols.

All initiators were successful in triggering the thiol-ene coupling for DUB labelling, using both recombinant purified proteins and complex cellular mixtures. Considering the results obtained in the biological settings tested, all three initiators are suitable for DUB labelling using the HA-$^{1-75}$Ub-alkene. If one preferential method were to be selected, Mn(OAc)$_3$ would be the most recommended for purified recombinant proteins, as it affords the strongest labelling of the three. Due to the high concentration of Mn(OAc)$_3$ required in more complex cellular settings, Irgacure 2959 would be the recommended method for assays using cell lysates, as it generates very strong labelling in a short period of time without the need for a degassing step. Irgacure 2959 was preferred over Mes-Acr$^+$ since protein degradation was observed at high concentrations of latter and longer blue light irradiation times. Though both UV and blue light may have a damaging effect on proteins[56–59], no degradation was observed using UV and Irgacure 2959. These results validate the versatility of thiol-ene coupling activation in both simple and complex protein settings and demonstrate its broad potential for use in protein profiling.

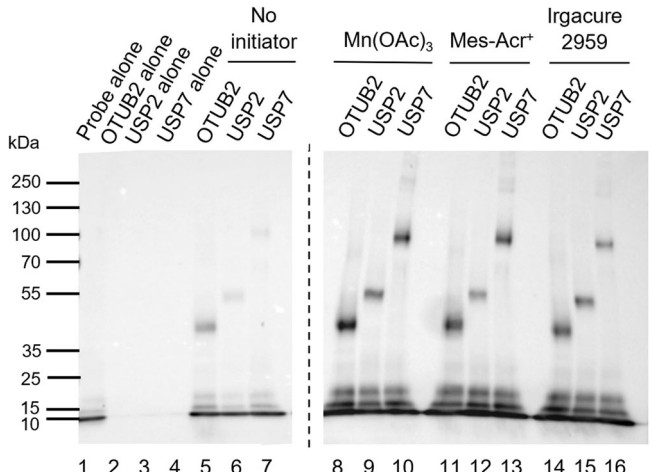

**Fig. 7 | Anti-HA Western Blot analysis of the labelling of selected recombinant DUBs (OTUB2, USP2 and USP7) with HA-$^{1-75}$Ub-alkene probe 1.** HA-$^{1-75}$Ub-alkene probe **1** (2 μg) was incubated with recombinant DUBs OTUB2 (1 μg), USP2 (2 μg) and USP7 (5 μg), for 90 min at 37 °C, prior to the addition of Irgacure 2959 (250 μM, 2-min UV irradiation), Mes-Acr$^+$ (100 μM, 10-min blue light irradiation) or Mn(OAc)$_3$ (125 μM, 30-min incubation at 37 °C). A negative control without initiator was performed. Results were analysed by SDS-PAGE and visualised by anti-HA Western Blot.

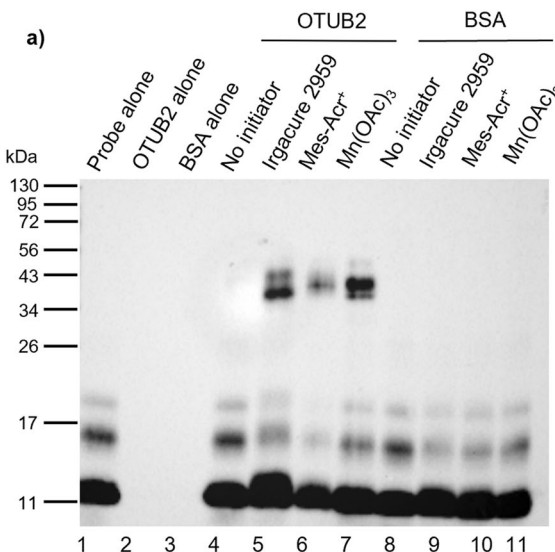

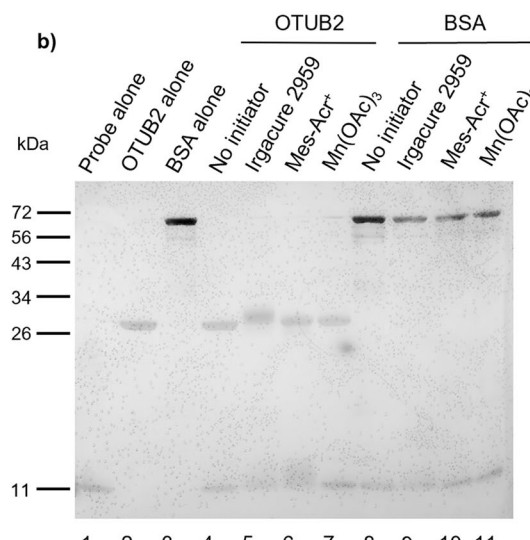

**Fig. 8 | Analysis of the labelling of recombinant DUB OTUB2 and non-DUB protein BSA, with HA-$^{1-75}$Ub-alkene probe 1.** HA-$^{1-75}$Ub-alkene probe **1** (2 μg) was incubated with the recombinant OTUB2 (1 μg) and BSA (2.5 μg), for 90 minutes at 37 °C, prior to the addition of Irgacure 2959 (250 μM, 2-min UV irradiation), Mes-Acr$^+$ (100 μM, 10-min blue light irradiation) or Mn(OAc)$_3$ (125 μM, 30-min incubation at 37 °C). A negative control without initiator was performed for each protein. Results were analysed by SDS-PAGE. **a** Visualisation by anti-HA Western Blot, **b** Visualisation by Silver Stain used as a control for the loading of both proteins (recombinant OTUB2 and BSA).

## Methods

### Chemical models

**General.** All commercially available reagents and solvents were purchased from commercial suppliers Merck, TCI Europe and Fluorochem EU, and used without further purification.

Purification of all compounds was performed using pro analysis solvents. Thin-layer chromatography (TLC) analysis was conducted with aluminium plates pre-coated with 0.2 mm MerckTM KGaA 60 M silica gel, $F_{254}$, 20 × 20 cm, and analysed using a UVP UVGL-58 lamp, using wavelengths of 254 and 365 nm. When UV visualisation was not possible, staining with $KMnO_4$ or ninhydrin was used. Column chromatography was performed using MerckTM silica gel 60 (0.040–0.063 nm).

Nuclear magnetic resonance (NMR) experiments were carried out using a Bruker 400 MHz system (Bruker, BioSpin, GmBH, Germany). Chemical shifts used the δ scale in parts per million (ppm), referenced to tetramethylsilane as an internal standard. Coupling constants (J) were described in Hertz (Hz). Peak multiplicity was recorded as singlet (s), broad singlet (bs), doublet (d), doublet of doublets (dd), doublet of doublets of doublets (ddd), triplet (t), triplet of doublets (td), quartet (q), pentet (p) and multiplet (m). Infrared (IR) analysis was performed using a Perkin Elmer spectrophotometer. Melting points (mp) were measured in a Griffin melting points apparatus. ESI mass spectra were acquired using a Bruker microTOF-Q III spectrometer interfaced to a Dionex UltiMate 3000 LC in positive and negative modes.

**Synthesis of methyl N-(tert-butoxycarbonyl)-S-(3-hydroxypropyl) cysteinate (4).** N-Boc-cysteine methyl ester (2) (131 µL, 0.637 mmol) and allyl alcohol (3) (65 µL, 0.637 mmol) were mixed in the appropriate solvent (4 mL). The initiator (DPAP: 163 mg, 0.637 mmol; MAP: 96 mg, 0.637 mmol; Irgacure 2959: 143 mg, 0.637 mmol; Mes-Acr⁺: 262 mg, 0.637 mmol; manganese acetate: 171 mg, 0.637 mmol) was added and the mixture was degassed. For initiators DPAP-MAP and Irgacure 2959, the mixtures were irradiated with UV at 254 nm in a Luzchem LZC-4 UV Reactor (14 lamps of 8 W, 112 W) for 30 min. For initiator Mes-Acr⁺, the mixture was irradiated with a blue LED (90 W) for 30 min and 6 h. For $Mn(OAc)_3$, the reaction was stirred at room temperature for 30 min. TLC analysis showed consumption of the limiting reagent 2 ($R_f$ 0.57; hexane-ethyl acetate 7:3) and formation of the product 4 ($R_f$ 0.09; hexane-ethyl acetate 7:3). The crude was concentrated under reduced pressure and purified using column chromatography (elution with hexane-ethyl acetate, 9:1 to 4:6). The desired product was isolated as a colourless oil (DPAP-MAP: 120 mg, 0.409 mmol, 64%; Irgacure 2959: 154 mg, 0.532 mmol, 84%; Mes-Acr⁺: 6 mg, 0.02 mmol, 3%; $Mn(OAc)_3$: 5 mg, 0.017 mmol, 3%). ¹H NMR (400 MHz, CDCl₃) δ 5.40 (d, J = 8.4 Hz, 1H, Cys-NH), 4.52–4.47 (m, 1H, Cys-CH), 3.73 (s, 3H, Cys-OCH₃), 3.71–3.63 (m, 2H, CH₂-OH), 2.95 (dd, J = 14.0, 4.9 Hz, 1H, Cys-CH₂), 2.87 (dd, J = 14.0, 6.0 Hz, 1H, Cys-CH₂), 2.71–2.56 (m, 2H, S-CH₂-CH₂), 2.35 (bs, 1H, OH), 1.88–1.68 (m, 2H, CH₂-CH₂-CH₂), 1.41 (s, 9H, Boc-CH₃); ¹³C NMR (125 MHz, CDCl₃) δ 171.73 (Cys-C = O), 155.40 (Boc-C = O), 80.32 (Boc-Cq), 60.88 (CH₂-OH), 53.13 (Cys-CH), 52.64 (Cys-OCH₃), 34.77 (Cys-CH₂), 31.90 (CH₂-CH₂-CH₂), 29.06 (S-CH₂-CH₂), 28.35 (Boc-CH₃); IR (neat) $v_{max}$: 3379, 2933, 1693, 1504, 1366, 1248, 1215, 1160, 1051, 1015, 732 cm⁻¹; HRMS-ESI (m/z): $[M+Na]^+$ calcd. for $C_{12}H_{23}NO_5S$, 316.1189; found, 316.1188.

**Characterisation of dimethyl 3,3'-disulfanediylbis(2-((tert-butoxycarbonyl)amino) propanoate) (5).** Characterisation of the product is in accordance with the literature[60,61]. $R_f$ 0.28, hexane-ethyl acetate 7:3; mp 91-93°C; ¹H NMR (400 MHz, CDCl₃) δ 5.39 (d, J = 8.2 Hz, 2H, Cys-NH), 4.60–4.55 (m, 2H, Cys-CH), 3.74 (s, 6H, Cys-OCH₃), 3.14 (d, J = 5.3 Hz, 4H, Cys-CH₂), 1.42 (s, 18H, Boc-CH₃); ¹³C NMR (125 MHz, CDCl₃) δ 171.28 (Cys-C = O), 155.15 (Boc-C = O), 80.37 (Boc-Cq), 52.89 (Cys-CH), 52.72 (Cys-OCH₃), 41.35 (Cys-CH₂), 28.38 (Boc-CH₃); IR (neat) $v_{max}$: 3379, 2933, 1745, 1683, 1505, 1360, 1211, 1159, 1056, 1019,

782 cm⁻¹; HRMS-ESI (m/z): $[M+Cl]^-$ calcd. for $C_{18}H_{32}N_2O_8S_2$, 503.1294; found, 503.1274.

### Labelling studies

**Expression and purification of HA-¹⁻⁷⁵Ub-MESNa.** BL21 (DE3) cells containing a pTYB2 plasmid encoding for a HA-tagged ⁷⁵Ub fusion protein with an intein domain and chitin-binding domain (HA-¹⁻⁷⁵Ub-intein-CBD) were cultured in LB medium (8 mL), containing ampicillin (100 µg/mL), at 37 °C, 180 rpm, for 18 h. The resulting culture was transferred into fresh LB medium (300 mL), containing ampicillin (100 µg/mL), and grown for an additional 2.5 h, at 37 °C, 180 rpm, up to an $OD_{600}$ 0.6–0.9. IPTG was then added at a final concentration of 0.4 mM and the bacteria were incubated for 20 h, at 18 °C, 180 rpm. The cell culture was centrifuged at 6000 rpm, for 15 min, and the resulting pellet was re-suspended in column buffer (50 mM HEPES pH 6.8, 100 mM NaOAc). Cell lysis was performed using a sonication tip, for 5 min with a 3-s pulse. The lysate mixture was centrifuged for 45 min, at 13,000 rpm. Chitin resin (5 mL) (New England Biolabs) was added to a column and the system was equilibrated using column buffer (50 mL). The supernatant was added and the column was washed using column buffer (50 mL). Column buffer containing MESNa (15 mL, 50 mM) was then run through the column and the system was incubated at 37 °C, for 18 h, with gentle shaking. HA-¹⁻⁷⁵Ub-MESNa was eluted with column buffer (8 mL) and the sample concentrated by centrifugation at 11,000 rpm in 5 kDa MW cut-off Vivaspin 500 centrifugal concentrators (Sartorious, Göttingen Germany), until a volume of 500 µL. HA-¹⁻⁷⁵Ub-MESNa was loaded into a NAP-5 column (GE Healthcare, Illinois USA) and desalted with column buffer (1 mL). Final protein concentration was determined using Nanodrop (3.2 mg/mL, 1 mL)[12,62,63].

**Synthesis of HA-¹⁻⁷⁵Ub-alkene probe 1.** A solution of N-Hydroxysuccinimide (0.2 M, 45 µL) and Tris-Cl pH 7.5 (100 mM, 10 µL) was added to HA-¹⁻⁷⁵Ub-MESNA in column buffer (3.2 mg/mL, 500 µL) and the mixture was incubated at 37 °C, for 10 min, with gentle shaking. Allylamine (15 µL, 0.2 mmol) was dissolved in a solution of MeCN-dH₂O (1:1, 50 µL) and added to the reaction mixture. The pH was adjusted to 10. The reaction was incubated at 37 °C, for 20 h, with gentle shaking. After this incubation, the reaction mixture was loaded into a NAP-5 column, desalted with column buffer (1 mL) and concentrated by spinning at 11,000 rom in a 5 kDa MW cut-off Vivaspin centrifugal concentrator. The protein concentration was determined using a BCA assay (4.7 mg/mL, 100 µL)[12].

**In vitro thiol-ene labelling of recombinant DUBs and HEK293T cell lysates with HA-¹⁻⁷⁵Ub-alkene probe1.** Probe 1 (2 µg) was incubated with recombinant DUBs (OTUB1 (1 µg), OTUB2 (Sinobiological, 13177-H07E) (1 µg), USP2 (Bio-techne, E-504-050) (1.5 µg) and USP7 (Sinobiological, 11681-H20B) (4 µg)), BSA (Merck Life Sciences Limited, A3059) (2.5 µg), β-galactosidase (Sigma-Aldrich, G5160) (5 µg) or HEK293T cell lysates (50 µg). The final volume was adjusted to 31 µL with homogenate buffer (50 mM Tris-Cl pH 7.4, 5 mM MgCl₂, 250 mM sucrose). The mixtures were pre-incubated for 90 min, at 37 °C, with gentle shaking, before the addition of initiators. For DPAP/MAP initiation, DPAP (0.5 mM) and MAP (0.5 mM) were added to the system and following degassing, the mixture was irradiated with UV at 254 nm in a Luzchem LZC-4 UV Reactor (14 lamps of 8 W, 112 W) for 2 min. Variable concentrations and irradiation/incubation times were attempted for Irgacure 2959 (concentration range 5 µM–5 mM; irradiation times 0.5–5 min), Mes-Acr⁺ (concentration range 5 µM–5 mM; irradiation times 0.5–15 min) and $Mn(OAc)_3$ (concentration range 2.5 µM–10 mM; incubation times 1–90 min). For Irgacure 2959 initiation, after optimisation, Irgacure 2959 (0.25 mM) was added to the mixture and the solution was irradiated with UV at 254 nm in a Luzchem LZC-4 UV Reactor (14 lamps of 8 W, 112 W) for 2 min. For Mes-Acr⁺,

following optimisation, Mes-Acr$^+$ (0.1 mM) was added to the system and the mixture was irradiated with a blue LED (36 W) for 10 min or with a blue LED (90 W) for 2 min. For Mn(OAc)$_3$, after optimisation, the initiator (5 mM) was added to the mixture and the solution was incubated for 60 min at 37 °C with gentle shaking. Upon completion, reducing sample buffer (31 µL) (0.2 M Tris-Cl pH 6.8, 30% glycerol, 0.4% β-mercaptoethanol, 9% SDS, 0.1% bromophenol blue) was added and the proteins were denatured at 95 °C for 5 min. Labelling was analysed using SDS-PAGE and visualised using silver anti-HA western blotting.

## Data availability

All data is available within the article and in the Supplementary Data provided. Uncropped Western Blots and SDS-PAGE are provided in Supplementary Fig. S6. NMR spectra are provided in Supplementary Fig. S7.

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

## Acknowledgements
This project was funded by the Higher Education Authority (HEA), project 213057 (A.C., D.L., T.H., and J.F.M.) and through funding from Science Foundation Ireland, 22/FFP-P/11234 (A.C. and J.F.M.), and 22/FFP-A/10459 (M.B. and A.G.B.).

## Author contributions
Conceptualisation, A.C. and J.F.M.; funding acquisition, J.F.M., D.B.L., and T.H.; methodology, A.C., J.F.M., D.B.L., and T.H.; experimental data collection and analysis, A.C.; cell culture, M.B. and A.G.B.; supervision, J.F.M.; project administration, J.F.M. and D.B.L.; writing, A.C. and J.F.M.

## Competing interests
The authors declare no competing interests.
