## [Transparent Peer Review file · Communications Chemistry]

Chemical- and photo-activation of protein-protein thiol-ene coupling for protein profiling

Corresponding Author: Professor Joanna McGouran

Version 0:

Reviewer comments:

Reviewer #1

(Remarks to the Author)

Authors provided a detailed study about the optimization of protein-protein thiol-ene coupling using various chemical activations. Various models were considered from simple solution chemistry to complex cell lysates. The manuscript is well written and importantly provides the adequate controls enabling to assess the comparative performance of the various methods.

By analyzing various experimental conditions for each method, clear guidelines are presented to improve their efficacy. Several options are introduced to activate protein-protein coupling with an increased efficacy compared to the previously described method used here as positive control.

The manuscript may thus be accepted after few minor revisions.

The main question is about the applicability range of this technique. This study clearly shows that efficiency, even without degassing and the selectivity of the coupling is closely related to the protein-protein pre-interaction and the pre-alignment of the thiol and alkene reactive groups in the DUB binding pocket. This should be further discussed at the end of the manuscript.

Another question is about the possible side effects on the biological samples accompanying each of the methods. The authors mentioned at several occasions that they observed protein degradation. This aspect should also be further discussed mentioning what are the possible side reactions and how to limit them.

This can be used to indicate which method/experimental conditions is the most recommended in conclusion.

Other comments:

- It would be useful to provide the structure of the activators and their working principle in Supporting Information. $Mn(OAc)_3$ activator is it really an initiator?
- The conditions of illumination (lamp type, set-up...) are not well-described in the Experimental Part for the preliminary studies with chemical models (line 394). Can they be really compared to the ones used with the cell lysates?
- The difference between "Yield 4" and "Conversion 2 into 4" in Table 1 (line 105-108 in the text) is unclear. It needs further explanations
- Line 165: "(Lanes 4-7, Figure 3a)" is to be changed to "(Lanes 5-7, Figure 3a)"?
- Could a possible lower supply of oxygen inside the protein binding pocket explain the efficacy and selectivity of the coupling?
- Other control than BSA would be necessary to fully assess the selectivity of the coupling. Those results actually show that such methods would be restricted to very specific applications.

Reviewer #2

(Remarks to the Author)

This manuscript presents a systematic investigation of the reaction conditions for "chemical and photo activation of thio-ene coupling for protein labeling." The authors have previously worked on deubiquitinating enzyme (DUB) profiling using similar photo-redox chemistry. In this study, they focused on optimizing the redox catalyst and light source for the most effective cross-linking. The investigation was conducted systematically, from in vitro to cellular conditions, and the results provide valuable insights for the field of photo-redox protein labeling. Based on the study's merits, I agree that this manuscript is suitable for publication. However, several points should be addressed, as detailed below:

1. Despite preliminary studies on chemical models demonstrating the highest crosslinking yields (Irgacure 2959 with UV in

ethanol for 30 min: 84%, Mes-Acr with blue light for 6 hours: 80%), the authors initiated OTUB1 labeling using $Mn(OAc)_3$. While prior experience with this chemistry may explain this choice, the rationale behind this decision is unclear, especially given the superior yields observed with the other combinations. Why did the authors not use the conditions that achieved the highest crosslinking yields? Furthermore, Figures 6 and 7 show better crosslinking with $Mn(OAc)_3$ compared to Irgacure 2959 or Mes-Acr. It would be more informative if the authors could explain the apparent discrepancy between the model chemical reaction and the protein labeling conditions.

2. In Figure 7, the silver-stained gel used as a control for protein loading does not appear to show equal loading amounts across the samples. This inconsistency should be addressed to ensure accurate interpretation of the results.

Reviewer #3

(Remarks to the Author)

In this manuscript, McGouran and coworkers reported their detailed studies on protein-protein thiol-ene coupling. Various of radical reaction conditions, including chemical radical initiator, UV induced radical initiator, Visible light mediated photocatalyst, and chemical redox activation, were used in the studies. The work systematically studied all conditions in model thiol-ene reaction between a protected cysteine and allyl alcohol and in protein-probe interaction setting. Although the information obtained in the model reaction is not 100% useful in the protein-protein coupling, such systematically study provided extremely useful information for the area of thiol-ene coupling of protein and other biologics. Thus, I recommend this manuscript publish in Communications chemistry after minor revision.

1. In the carton of Figure 1b, radical thiol-ene reaction prefers to provide non-Markovnikov thiol-ene adduct, so the "S" on DUB should terminal carbon of the alkene warhead.

2. Page 3, line 5. "The thiol-ene product 4 was obtained with DPAP and MAP, with an isolated yield of 63%". The yield is not the same with the one, 64%, in Table 1.

3. It would useful to list the chemical structures of Irgacure 2959 and Mes-Acr+.

Version 1:

Reviewer comments:

Reviewer #1

(Remarks to the Author)

The points previously raised have been satisfactorily addressed by the authors. The manuscript may thus be accepted for publication

Reviewer #2

(Remarks to the Author)

This revised manuscript addressed all issues raised by this reviewer properly. Thus this reviewer suggest accepting this manuscript in this form. Great piece of work!

Reviewer #3

(Remarks to the Author)

This reviewer is satisfied with the revision.

Trinity College Dublin
Coláiste na Tríonóide, Baile Átha Cliath
The University of Dublin

Prof. Joanna McGouran
Associate Professor
School of Chemistry
Trinity College Dublin
04/11/2024

Re: Manuscript ID: COMMSCHEM-24-0434

Dear Reviewers,

Thank you for your helpful comments on our manuscript by André Campanico and co-authors entitled "Chemical- and photo-activation of protein-protein thiol-ene coupling for protein profiling". We were pleased to hear that Reviewer 1 felt the manuscript "provided a detailed study", Reviewer 2 believes our "results provide valuable insights for the field of photo-redox protein labeling", and Reviewer 3 felt our manuscript "provided extremely useful information for the area of thiol-ene coupling of protein and other biologics".

We thank the reviewers for the time taken to carefully examine the manuscript and their helpful comments which we have incorporated into our revised manuscript. We have also prepared a detailed point-by-point reply in which we address the reviewers' comments. We feel this has served to strengthen the manuscript and we would now be delighted for it to be considered for publication in Communications Chemistry.

Addressing a request shared by both reviewers 2 and 3, we included a figure detailing the initiator structures (Figure 2). Following the reviewers' suggestions, we have included a detailed discussion of protein degradation under irradiation/radical conditions and we included further discussion of the silver stain results obtained. Furthermore, we included second negative control protein to further demonstrate the selectivity of the labelling reaction (β -galactosidase). In our point-by-point reply, we have also provided a point-by-point response to all of the reviewers queries.

With these additions, we do hope that our revised manuscript will be considered for publication in Communications Chemistry, and we look forward to hearing from you soon.

Yours sincerely,

Joanna McGouran
Associate Professor in Translational Organic Chemistry

Reviewer 1:

Authors provided a detailed study about the optimization of protein-protein thiol-ene coupling using various chemical activations. Various models were considered from simple solution chemistry to complex cell lysates. The manuscript is well written and importantly provides the adequate controls enabling to assess the comparative performance of the various methods.

By analyzing various experimental conditions for each method, clear guidelines are presented to improve their efficacy. Several options are introduced to activate protein-protein coupling with an increased efficacy compared to the previously described method used here as positive control.

The manuscript may thus be accepted after few minor revisions.

We thank the reviewer for their positive comments.

The main question is about the applicability range of this technique. This study clearly shows that efficiency, even without degassing and the selectivity of the coupling is closely related to the protein-protein pre-interaction and the pre-alignment of the thiol and alkene reactive groups in the DUB binding pocket. This should be further discussed at the end of the manuscript.

We agree with the reviewer that this is an important feature of the work. We see the requirement for a pre-interaction and pre-alignment as an advantage, as this requirement increases the specificity of the probe to its target and limits side reactivity. To reflect this, the Conclusion section has been modified from:

“Furthermore, the alignment between the alkene and thiol induced by protein-probe binding prior to activation overcomes the need for an oxygen-free environment, a disadvantageous and limiting step in biological applications.’

To:

‘Furthermore, the alignment between the alkene and thiol induced by protein-probe binding prior to activation overcomes the need for an oxygen-free environment, a disadvantageous and limiting step in biological applications. The requirement for such a pre-interaction, which will align the alkene and thiol moieties for efficient thiol-ene reaction, also increases the specificity of the probe, limiting any side reactivity with other thiols.’

Another question is about the possible side effects on the biological samples accompanying each of the methods. The authors mentioned at several occasions that they observed protein degradation. This aspect should also be further discussed mentioning what are the possible side reactions and how to limit them.

We thank the reviewer for this comment. We have examined literature studies of the specific initiators, light sources and free-radical species on protein degradation.

The section ‘Initiator concentration dependence’ has been modified from:

‘The samples with Irgacure 2959 were irradiated with UV for 2 min, following the irradiation time previously optimised for DPAP/MAP¹². Although maximum labelling was obtained at the concentration of 1 mM (Lane 9, Figure 5a), significant lysate labelling can be observed using only 250 μM (Lane 7, Figure 5a).’

To:

'The samples with Irgacure 2959 were irradiated with UV for 2 min, following the irradiation time previously optimised for DPAP/MAP¹². Although maximum labelling was obtained at the concentration of 1 mM (Lane 9, Figure 5a), significant lysate labelling can be observed using only 250 μ M (Lane 7, Figure 5a) and that was selected as the optimal concentration, in an attempt to limit excess radical formation. Radical formation may lead to changes in the protein backbone, inducing protein denaturation and degradation⁵¹.

The following *reference* was added to the manuscript:

51 Xu, H., *et al.* Effects of Irgacure 2959 and lithium phenyl-2,4,6-trimethylbenzoylphosphinate on cell viability, physical properties, and microstructure in 3D bioprinting of vascular-like constructs. *Biomed Mater* 15, 055021 (2020). <https://doi.org/10.1088/1748-605X/ab954e>

The following was added to the section 'Initiator incubation time':

'For the samples with Irgacure 2959, optimal labelling was observed after 2 minutes of UV irradiation (Lane 7, Figure 6a). UV irradiation is known to promote irreversible protein denaturation, due to formation of free radicals and radical oxygen species leading to changes in primary, secondary and tertiary structures^{56,57}. The more susceptible residues to UV light excitation and free radical formation are tryptophan, tyrosine and phenylalanine, due to their aromaticity^{56,58}. Though there is no significant difference in DUB labelling between irradiating for 2 minutes or 5 minutes, a 2-minute irradiation was selected as optimal to shorten UV irradiation time and prevent protein denaturation and degradation.'

The following references were added to the manuscript:

56 Kerwin, B. A. & Remmele, R. L. *Protect from light: photodegradation and protein biologics. J Pharm Sci* 96, 1468-1479 (2007). <https://doi.org/10.1002/jps.20815>

57 Ruzza, P., *et al.* Free Radicals and ROS Induce Protein Denaturation by UV Photostability Assay. *Int J Mol Sci* 22, 6512 (2021). <https://doi.org/10.3390/ijms22126512>

58 Neves-Petersen, M. T., *et al.* Flash photolysis of cutinase: identification and decay kinetics of transient intermediates formed upon UV excitation of aromatic residues. *Biophys J* 97, 211-226 (2009). <https://doi.org/10.1016/j.bpj.2009.01.065>

The section 'Initiator incubation time' has been modified from:

'In this setting, an optimal irradiation time of 2 minutes was observed (Lane 7, Figure 6c), however, protein degradation was observed after 5 minutes (Lane 8-10, Figure 6c). These results demonstrate that quick, efficient thiol-ene labelling can be obtained with Mes-Acr⁺ and Irgacure 2959 in complex biological samples.'

To:

'In this setting, an optimal irradiation time of 2 minutes was observed (Lane 7, Figure 6c), however, protein degradation was observed after 5 minutes (Lane 8-10, Figure 6c). Blue light irradiation promotes oxidation of proteins through generation of superoxide, singlet oxygen species and hydrogen peroxide. The residues more susceptible to oxidation are cysteine, tyrosine, methionine, histidine and tryptophan⁵⁹. These modifications may have contributed to protein denaturation and degradation at longer irradiation times. Still, these results demonstrate that quick, efficient thiol-ene labelling can be obtained with Mes-Acr⁺ and Irgacure 2959 in complex biological samples.'

The following reference was added:

The section 'Initiator concentration dependence' has been modified from:

'Discrepancies between the incubation of probe **1** with OTUB1 and HEK293T cell lysates were observed, as 125 μM of $\text{Mn}(\text{OAc})_3$ was enough to label OTUB1 and a 40-fold concentration increase was required for the lysates. These were attributed to potential manganese side reactivity in the complex cellular mixture, which may limit the use of $\text{Mn}(\text{OAc})_3$ as an initiator in a complex cellular milieu.'

To:

'Discrepancies between the incubation of probe **1** with OTUB1 and HEK293T cell lysates were observed, as 125 μM of $\text{Mn}(\text{OAc})_3$ was enough to label OTUB1 and a 40-fold concentration increase was required for the lysates. Manganese (III) is known to interact with carbonyl moieties, inducing α -carbon deprotonation and enol formation, and consequent radical formation at the α -carbon⁵²⁻⁵⁵. Its reactivity has been demonstrated with ketones, carboxylic acids, esters and amides, suggesting that manganese (III) may interact with several biological macromolecules, including proteins. Despite its potential toxicity, protein degradation was not observed in any of the concentrations tested. However, this potential manganese side reactivity in complex cellular mixtures may limit the use of $\text{Mn}(\text{OAc})_3$ as an initiator in a complex cellular milieu.'

The following references were added:

52 Reaney, S. H., Kwik-Urbe, C. L. & Smith, D. R. Manganese oxidation state and its implications for toxicity. *Chem Res Toxicol* 15, 1119-1126 (2002). <https://doi.org/10.1021/tx025525e>

53 Bouhlel, A., et al. Manganese(III) acetate-mediated oxidative cyclization of α -methylstyrene and trans-stilbene with *b*-ketosulfones. *Molecules* 18, 4293-4307 (2013). <https://doi.org/10.3390/molecules18044293>

54 Liu, Z., et al. Unexpected manganese(III) acetate-mediated reactions of β -enamino carbonyl compounds with 1-(pyridin-2-yl)-enones under mechanical milling conditions. *Chem Commun* 48, 11665-11667 (2012). <https://doi.org/10.1039/C2CC36360G>

55 Hulcoop, D. G., et al. Manganese(III) acetate mediated synthesis of oxygen heterocycles. Influence of copper(II) salts on product distribution. *Org Biomol Chem* 2, 965-967 (2004). <https://doi.org/10.1039/B402411G>

This can be used to indicate which method/experimental conditions is the most recommended in conclusion.

We agree with the reviewer that the 'Conclusion' section would benefit from a discussion comparing the initiators as to which is the most recommended method. Considering the results obtained in the manuscript, the following has been added to the section 'Conclusions':

'All initiators were successful in triggering the thiol-ene coupling for DUB labelling, using both recombinant purified proteins and complex cellular mixtures. Considering the results obtained in the biological settings tested, all three initiators are suitable for DUB labelling using the $\text{HA}^{-1-75}\text{Ub}$ -alkene. If one preferential method were to be selected, $\text{Mn}(\text{OAc})_3$ would be the most recommended for purified recombinant proteins, as it affords the strongest labelling of the three. Due to the high

concentration of $\text{Mn}(\text{OAc})_3$ required in more complex cellular settings, Irgacure 2959 would be the recommended method for assays using cell lysates, as it generates very strong labelling in a short period of time without the need for a degassing step. Irgacure 2959 was preferred over Mes-Acr⁺ since protein degradation was observed at high concentrations of latter and longer blue light irradiation times. Though both UV and blue light may have a damaging effect on proteins⁵⁶⁻⁵⁹, no degradation was observed using UV and Irgacure 2959. These results validate the versatility of thiol-ene coupling activation in both simple and complex protein settings and demonstrate its broad potential for use in protein profiling.'

It would be useful to provide the structure of the activators and their working principle in Supporting Information. $\text{Mn}(\text{OAc})_3$ activator is it really an initiator?

We thank the reviewer for this comment and agree that adding the structure and activating mechanism of the initiators would be useful. Regarding $\text{Mn}(\text{OAc})_3$, although it is not a photoinitiator, it initiates a similar reaction by activating the active-site thiol through a redox mechanism. Hence, we chose to use the term redox initiator to distinguish it from the light-dependent initiators whilst still making it clear that it activates the same reaction and generates the same product.

To accommodate the reviewer's suggestions, Figure 2 has been added:

'Figure 2. Initiators selected to trigger the thiol-ene coupling in this study: a) Irgacure 2959; b) 9-mesityl-10-methylacridinium perchlorate (Mes-Acr⁺); c) manganese (III) acetate.'

Supplementary Figure S1 was also added to the Supplementary Information. Supplementary references 3, 4 and 5 were also added.

- 3 Tomal, W. & Ortyl, J. *Water-Soluble Photoinitiators in Biomedical Applications. Polymers (Basel)* **12** (2020). <https://doi.org/10.3390/polym12051073>
- 4 Romero, N. A. & Nicewicz, D. A. *Mechanistic insight into the photoredox catalysis of anti-markovnikov alkene hydrofunctionalization reactions. J Am Chem Soc* **136**, 17024-17035 (2014). <https://doi.org/10.1021/ja506228u>
- 5 Snider, B. B. *Mechanisms of $\text{Mn}(\text{OAc})_3$ -based oxidative free-radical additions and cyclizations. Tetrahedron* **65**, 10735-10744 (2009). <https://doi.org/10.1016/j.tet.2009.09.025>

'Supplementary Figure S1. Initiation mechanism of the three initiators selected to trigger the thiol-ene coupling: a) Irgacure 2959³; b) 9-mesityl-10-methylacridinium perchlorate (Mes-Acr⁺)⁴; c) manganese (III) acetate⁵.'

In order to direct the reader to the new figures added, the Introduction section has been modified from:

'2-Hydroxy-4'-(2-hydroxyethoxy)-2-methylpropiophenone (Irgacure 2959)^{43,44} and 9-mesityl-10-methylacridinium perchlorate (Mes-Acr⁺)^{16,45} were selected as the light-induced radical initiators, due to their proven ability to catalyse the thiol-ene coupling in peptides. Irgacure 2959 and Mes-Acr⁺ are triggered by UV and blue wavelengths, respectively. In an attempt to induce the thiol-ene coupling through a redox mechanism, manganese acetate was selected as the redox initiator, due to its previously demonstrated ability to activate thiols⁴⁶.'

To:

'2-Hydroxy-4'-(2-hydroxyethoxy)-2-methylpropiophenone (Irgacure 2959)^{43,44} and 9-mesityl-10-methylacridinium perchlorate (Mes-Acr⁺)^{16,45} were selected as the light-induced radical initiators (**Figure 2a-b, Figure S1a-b**), due to their proven ability to catalyse the thiol-ene coupling in

peptides. Irgacure 2959 and Mes-Acr⁺ are triggered by UV and blue wavelengths, respectively. In an attempt to induce the thiol-ene coupling through a redox mechanism, manganese acetate was selected as the redox initiator (Figure 2c, Figure S1c), due to its previously demonstrated ability to activate thiols⁴⁶.

The conditions of illumination (lamp type, set-up...) are not well-described in the Experimental Part for the preliminary studies with chemical models (line 394). Can they be really compared to the ones used with the cell lysates?

The lamps and set-up that were used for the chemical models were the same as the ones used for the cell lysates. The following information was added to the Experimental section:

'For initiators DPAP-MAP and Irgacure 2959, the mixtures were irradiated with UV at 254 nm in a Luzchem LZC-4 UV Reactor (14 lamps of 8W, 112W) for 30 min. For initiator Mes-Acr⁺, the mixture was irradiated with a blue LED (90 W) for 30 min and 6 hours. For Mn(OAc)₃, the reaction was stirred at room temperature for 30 min.'

The difference between "Yield 4" and "Conversion 2 into 4" in Table 1 (line 105-108 in the text) is unclear. It needs further explanations.

The yield presented refers to the isolated product yield calculated from the amount of product formed in the reaction, while for the calculated conversion, the amount of starting material that was recovered was subtracted and only the starting material that was consumed in the reaction was taken into consideration in calculating the conversion.

To clarify this, the following sentence has been reworded from:

'Although the isolated yield was low, the conversion of **2** into **4**, calculated as the percentage of starting material consumed in the reaction that converted into the desired thiol-ene product, was 69%.'

to:

'Although the isolated yield was low, 95% of starting material **2** was recovered. From the 5% starting material consumed in the reaction the isolated yield of 3% of **4** represents a 69% conversion of the consumed starting material into desired product.'

A column with the percentage of starting material **2** has also been added to Table 1. Table 1 was then modified from:

Entry	Initiator	Light source	Solvent	Degassing	Reaction time	Yield 4	Conversion 2 into 4	Disulfide Yield
1	-	-	EtOH	No	30 min	NR	-	-
2	-	-	EtOH	Yes	30 min	NR	-	-
3	DPAP/MAP	UV	EtOH	Yes	30 min	64%	64%	-
4	Irgacure 2959	UV	EtOH	No	30 min	18%	18%	-
5	Irgacure 2959	UV	EtOH	Yes	30 min	84%	84%	-
6	Mes-Acr ⁺	Blue Light	EtOH	No	30 min	NR	-	-
7	Mes-Acr ⁺	Blue Light	ACN	No	30 min	NR	-	-
8	Mes-Acr ⁺	Blue Light	ACN	Yes	30 min	3%	69%	-
9	Mes-Acr ⁺	Blue Light	ACN	Yes	6 hours	12%	80%	-
10	Mn(OAc) ₃	-	EtOH	No	30 min	NR*	-	58%
11	Mn(OAc) ₃	-	EtOH	Yes	30 min	3%*	3%	46%

*disulfide formation was observed; NR – no reaction

To:

Entry	Initiator	Light source	Solvent	Degassing	Reaction time	Yield 4	Recovered 2	Conversion 2 into 4	Disulfide Yield
1	-	-	EtOH	No	30 min	NR	∓	-	-
2	-	-	EtOH	Yes	30 min	NR	∓	-	-
3	DPAP/MAP	UV	EtOH	Yes	30 min	64%	∓	64%	-
4	Irgacure 2959	UV	EtOH	No	30 min	18%	∓	18%	-
5	Irgacure 2959	UV	EtOH	Yes	30 min	84%	∓	84%	-
6	Mes-Acr ⁺	Blue Light	EtOH	No	30 min	NR	∓	-	-
7	Mes-Acr ⁺	Blue Light	ACN	No	30 min	NR	∓	-	-
8	Mes-Acr ⁺	Blue Light	ACN	Yes	30 min	3%	95%	69%	-
9	Mes-Acr ⁺	Blue Light	ACN	Yes	6 hours	12%	85%	80%	-
10	Mn(OAc) ₃	-	EtOH	No	30 min	NR*	∓	-	58%
11	Mn(OAc) ₃	-	EtOH	Yes	30 min	3%*	∓	3%	46%

*disulfide formation was observed; NR – no reaction

Line 165: “(Lanes 4-7, Figure 3a)” is to be changed to “(Lanes 5-7, Figure 3a)”?

We agree that this should be clarified to highlight which samples were irradiated and which ones were kept in the dark. To make this clearer, the following sentence was modified from:

‘When the reaction mixtures were kept in the dark, no labelling was observed (Lanes 4-7, Figure 4a), while light-irradiated samples afforded efficient thiol-ene labelling.’

to:

‘When the reaction mixtures were kept in the dark, no labelling was observed (Lane 5 and 7, Figure 4a), while light-irradiated samples afforded efficient thiol-ene labelling (Lanes 4 and 6, Figure 4a).’

Could a possible lower supply of oxygen inside the protein binding pocket explain the efficacy and selectivity of the coupling?

Although this would be a plausible explanation for the results obtained in non-degassed samples, this hypothesis is not supported by previous results obtained in the group (Taylor, N. C., Hessman, G., Kramer, H. B. & McGouran, J. F. *Probing enzymatic activity - a radical approach. Chem. Sci.* **11**, 2967-2972 (2020)). When the thiol-ene reaction was triggered using a mixture of DPAP/MAP, labelling was not observed in non-degassed samples, suggesting that this phenomenon is initiator dependent.

To further clarify this issue, section 'Role of light-irradiation and degassing' was modified from:

'It was however observed that, unlike in previous studies with DPAP/MAP, it was not essential for the thiol-ene coupling, as significant thiol-ene labelling was still observed in the non-degassed sample.'

To:

'It was however observed that, unlike in previous studies with DPAP/MAP, it was not essential for the thiol-ene coupling, as significant thiol-ene labelling was still observed in the non-degassed sample. This result suggests that the need for an oxygen-free environment might be initiator-dependent.

Other control than BSA would be necessary to fully assess the selectivity of the coupling. Those results actually show that such methods would be restricted to very specific applications.

To further confirm the selectivity of the thiol-ene coupling, an additional negative control experiment was performed with β -galactosidase. HA-¹⁻⁷⁵Ub-alkene probe 1 was incubated with the DUB USP7 and the non-DUB β -galactosidase. Similar to the assay with BSA, labelling was only observed with the selected DUB.

To include this assay, section 'Incubation of probe 1 and recombinant DUB and non-DUB models' was modified from:

'A set of final assays were performed using the optimised conditions for each initiator with three recombinant DUBs (OTUB2, USP2 and USP7) and the non-DUB BSA. (...) Gratifyingly, labelling was not observed using BSA (lanes 9-11, Figure 8a), even though a larger amount of the non-DUB protein was used in the assay (2.5 μ g of BSA vs 1 μ g of OTUB2). The inability of probe 1 to label BSA supports probe selectivity for DUBs.'

To:

'A set of final assays were performed using the optimised conditions for each initiator with three recombinant DUBs (OTUB2, USP2 and USP7) and the non-DUBs BSA and β -galactosidase. (...) Gratifyingly, labelling was not observed using BSA (lanes 9-11, Figure 8a), even though a larger amount of the non-DUB protein was used in the assay (2.5 μ g of BSA vs 1 μ g of OTUB2). The same result was obtained with β -galactosidase, with labelling observed only with the DUB used (lanes 5-7 vs lanes 9-11, Figure S5). The inability of probe 1 to label BSA and β -galactosidase supports probe selectivity for DUBs.'

Figure S5 was also added to the Supplementary Information:

Supplementary Figure S5. Anti-HA Western Blot analysis of the incubation of the HA-¹⁻⁷⁵Ub-alkene probe 1 with USP7 and β -galactosidase. HA-¹⁻⁷⁵Ub-alkene probe 1 (2 μ g) was incubated with the recombinant USP7 (4 μ g) and β -galactosidase (5 μ g), for 90 minutes at 37 °C, prior to the addition of Irgacure 2959 (250 μ M, 2-min UV irradiation), Mes-Acr⁺ (100 μ M, 10-min blue light irradiation) or Mn(OAc)₃ (125 μ M, 30-min incubation at 37 °C). A negative control without initiator was performed for each protein. Results were analysed by SDS-PAGE. a) Visualisation by anti-HA Western Blot, b)

Visualisation by Silver Stain used as a control for the loading of both proteins (recombinant USP7 and β -galactosidase).

To accommodate this change, the 'Introduction' section has been modified from:

'Following protein-protein thiol-ene optimisation, probe **1** was then incubated with different recombinant DUB models and with the non-DUB protein bovine serum albumine (BSA) to demonstrate the selectivity and pre-binding requirement for labelling under these conditions. For all initiators, thiol-ene was successful with the selected recombinant DUBs, but not with BSA, supporting the selectivity of this system.'

To:

'Following protein-protein thiol-ene optimisation, probe **1** was then incubated with different recombinant DUB models and with the non-DUB proteins bovine serum albumine (BSA) and β -galactosidase to demonstrate the selectivity and pre-binding requirement for labelling under these conditions. For all initiators, thiol-ene was successful with the selected recombinant DUBs, but not with BSA and β -galactosidase, supporting the selectivity of this system.'

The 'Experimental' section has also been modified from:

'Probe **1** (2 μ g) was incubated with recombinant DUBs (OTUB1 (1 μ g), OTUB2 (Sinobiological, 13177-H07E) (1 μ g), USP2 (Bio-technie, E-504-050) (1.5 μ g) and USP7 (Sinobiological, 11681-H20B) (4 μ g)), BSA (Merck Life Sciences Limited, A3059) (2.5 μ g) or HEK293T cell lysates (50 μ g).'

To:

'Probe **1** (2 μ g) was incubated with recombinant DUBs (OTUB1 (1 μ g), OTUB2 (Sinobiological, 13177-H07E) (1 μ g), USP2 (Bio-technie, E-504-050) (1.5 μ g) and USP7 (Sinobiological, 11681-H20B) (4 μ g)), BSA (Merck Life Sciences Limited, A3059) (2.5 μ g), β -galactosidase (Sigma-Aldrich, G5160) (5 μ g) or HEK293T cell lysates (50 μ g).'

Reviewer 2:

This manuscript presents a systematic investigation of the reaction conditions for “chemical and photo activation of thio-ene coupling for protein labeling.” The authors have previously worked on deubiquitinating enzyme (DUB) profiling using similar photo-redox chemistry. In this study, they focused on optimizing the redox catalyst and light source for the most effective cross-linking. The investigation was conducted systematically, from in vitro to cellular conditions, and the results provide valuable insights for the field of photo-redox protein labelling. Based on the study’s merits, I agree that this manuscript is suitable for publication. However, several points should be addressed, as detailed below:

We thank the reviewer for their positive comments.

1. Despite preliminary studies on chemical models demonstrating the highest crosslinking yields (Irgacure 2959 with UV in ethanol for 30 min: 84%, Mes-Acr with blue light for 6 hours: 80%), the authors initiated OTUB1 labelling using Mn(OAc)₃. While prior experience with this chemistry may explain this choice, the rationale behind this decision is unclear, especially given the superior yields observed with the other combinations. Why did the authors not use the conditions that achieved the highest crosslinking yields? Furthermore, Figures 6 and 7 show better crosslinking with Mn(OAc)₃ compared to Irgacure 2959 or Mes-Acr. It would be more informative if the authors could explain the apparent discrepancy between the model chemical reaction and the protein labelling conditions.

The promising chemical model results with Irgacure 2959 and Mes-Acr⁺ meant that they were “fast-tracked” straight into the more challenging cell lysate system. Mn(OAc)₃ was hampered in the chemical model studies by disulfide formation. To investigate if disulfide formation could be suppressed by inducing proximity between the thiol and the alkene, OTUB1 labelling was selected as an additional preliminary method to test redox initiation of thiol-ene coupling.

To further clarify this, section ‘Preliminary studies with chemical models’ has been modified from:

‘Following this study, Irgacure 2959 and Mes Acr⁺ were considered suitable for more complex biological models due to the high yield and good conversions observed in the model systems. The Mn(OAc)₃ study indicated that it could be used to catalyse the thiol-ene reaction provided disulfide formation could be suppressed. It was reasoned that a protein-protein thiol-ene reaction, in which there was a binding event prior to thiol-ene initiation, would favour the thiol-ene reaction over disulfide formation due to the induced proximity of the thiol and alkene. To test this hypothesis, we investigated a model thiol-ene reaction in which protein-protein binding could be exploited to favour the thiol-ene product over disulfide formation.’

To:

‘Following this study, Irgacure 2959 and Mes Acr⁺ were considered suitable for more complex biological models due to the high yield and good conversions observed in the model systems. The Mn(OAc)₃ study indicated that it could be used to catalyse the thiol-ene reaction provided disulfide formation could be suppressed. It was reasoned that a protein-protein thiol-ene reaction, in which there was a binding event prior to thiol-ene initiation, would favour the thiol-ene reaction over disulfide formation due to the induced proximity of the thiol and alkene. To test this hypothesis, we included an additional model system to investigate a protein-protein thiol-ene reaction using model

enzyme OTUB1. In this system, we anticipated that binding interactions prior to radical initiation could be exploited to favour the thiol-ene product over disulfide formation.'

The title of section 'Preliminary studies with Mn(OAc)₃ and OTUB1' has been modified to 'Additional studies with Mn(OAc)₃ and OTUB1' and its content has been modified from:

'Although Mn(OAc)₃ displayed limited efficiency in triggering the thiol-ene coupling in the chemical model system (Entry 11, Table 1), Mn(OAc)₃ has been shown to effectively promote thiol activation⁴, leading to disulfide formation in the model system. To further assess the potential of Mn(OAc)₃ as a thiol-ene initiator, a second model system was employed with the HA-¹⁻⁷⁵Ub-alkene probe **1** and the recombinant DUB OTUB1 (OTU Domain-containing ubiquitin aldehyde-binding protein 1), a system previously validated using DUBs with the thiol-ene coupling using DPAP and MAP⁵.'

To:

'Although Mn(OAc)₃ displayed limited efficiency in triggering the thiol-ene coupling in the chemical model system (Entry 11, Table 1), Mn(OAc)₃ has been shown to effectively promote thiol activation⁴, leading to disulfide formation in the model system. To assess if induced proximity between the thiol and the alkene may contribute to suppression of disulfide formation and efficient thiol-ene coupling, a second model system was employed with the HA-¹⁻⁷⁵Ub-alkene probe **1** and the recombinant DUB OTUB1 (OTU Domain-containing ubiquitin aldehyde-binding protein 1), a system previously validated using DUBs with the thiol-ene coupling using DPAP and MAP⁵.'

Regarding the conditions used in protein labelling with the different initiators, the irradiation times were selected based on our previous experience with DPAP/MAP (*Taylor, N. C., Hessman, G., Kramer, H. B. & McGouran, J. F. Probing enzymatic activity - a radical approach. Chem. Sci. 11, 2967-2972 (2020)*). Previous optimisations with DPAP/MAP had showed that the optimal irradiation time for that system is 2 minutes. Considering that Irgacure 2959 is also activated by UV radiation, the timepoints selected for this time optimisation assay in section 'Initiator incubation time' were chosen knowing that the optimal irradiation time would likely be similar to DPAP/MAP. Furthermore, in our previous study with DPAP/MAP, protein degradation was observed when samples were irradiated for more than 10 minutes. Regarding Mes-Acr⁺, considering that blue light is less energetic than UV radiation, we hypothesised that a slightly longer irradiation time would be necessary. This theory proved to be correct. Regarding Mn(OAc)₃, to the best of our knowledge, there is no precedence for its use in similar systems. The initial incubation time selected was then 30 minutes, the same as the one used for the chemical models.

Considering this, section 'Initiator Incubation Time' has been modified from:

'For the samples with Irgacure 2959, optimal labelling was observed after 2 minutes of UV irradiation (Lane 7, Figure 6a).'

To:

'For the samples with Irgacure 2959, irradiation times were selected based on previous studies with DPAP/MAP¹² and optimal labelling was observed after 2 minutes of UV irradiation (Lane 7, Figure 6a).'

To address the different labelling intensities between Mn(OAc)₃, Irgacure 2959 and Mes-Acr⁺ in Figures 7 and 8, section 'Incubation of probe 1 and recombinant DUB and non-DUB models' was modified from:

'However, all initiators were successful in triggering the thiol-ene coupling between probe 1 and the recombinant DUBs selected as significantly stronger labelling bands were seen under these conditions (lanes 8-16 vs lanes 5-7, Figure 7).'

To:

'However, all initiators were successful in triggering the thiol-ene coupling between probe 1 and the recombinant DUBs selected as significantly stronger labelling bands were seen under these conditions (lanes 8-16 vs lanes 5-7, Figure 7). Furthermore, stronger labelling was observed with Mn(OAc)₃ when compared to Irgacure 2959 and Mes-Acr⁺, suggesting that redox initiation might be more efficient in the labelling of purified recombinant proteins than light-dependent initiation.

2. In Figure 7, the silver-stained gel used as a control for protein loading does not appear to show equal loading amounts across the samples. This inconsistency should be addressed to ensure accurate interpretation of the results.

In the Silver Stain in Figure 7, 1 µg of OTUB2 and 2.5 µg of BSA were used in each sample. Different amounts between the two samples were used due to their different molecular weights (OTUB2: 27 kDa; BSA: 66.5 kDa). This discrepancy led to higher intensity of the BSA bands in the Silver Stain when compared to OTUB2.

Although the same amount of BSA was added to all samples, there is a difference in band intensity between the lane with no initiator (Figure 8b, lane 8) and the lanes with initiators (Figure 8b, lanes 9-11). This difference was attributed to protein degradation addressed in other parts of the manuscript.

To address the concern raised by the reviewer, the following text has been modified from:

'Gratifyingly, labelling was not observed using BSA (lanes 9-11, Figure 8a).'

To:

'Gratifyingly, labelling was not observed using BSA (lanes 9-11, Figure 8a), even though a larger amount of the non-DUB protein was used in the assay (2.5 µg of BSA vs 1 µg of OTUB2). The inability of probe 1 to label BSA supports probe selectivity for DUBs. A difference in band intensity was observed for BSA between the sample with no initiator (Figure 8b, lane 8) and the sample with the selected initiators (Figure 8b, lanes 9-11). This difference was attributed to protein degradation caused by the different conditions used to trigger the thiol-ene coupling.'

Reviewer 3:

In this manuscript, McGouran and coworkers reported their detailed studies on protein-protein thiol-ene coupling. Various of radical reaction conditions, including chemical radical initiator, UV induced radical initiator, Visible light mediated photocatalyst, and chemical redox activation, were used in the studies. The work systematically studied all conditions in model thiol-ene reaction between a protected cysteine and allyl alcohol and in protein-probe interaction setting. Although the information obtained in the model reaction is not 100% useful in the protein-protein coupling, such systematically study provided extremely useful information for the area of thiol-ene coupling of protein and other biologics. Thus, I recommend this manuscript publish in Communications chemistry after minor revision.

We thank the reviewer for their positive comments.

1. In the carton of Figure 1b, radical thiol-ene reaction prefers to provide non-markovnikov thiol-ene adduct, so the "S" on DUB should terminal carbon of the alkene warhead.

The regioselectivity of the proposed bond formation is based on analogy with previous work with ubiquitin electrophilic probes, which has shown that the active-site thiol preferentially reacts with C₂ position (Mons, E. et al. *Exploring the Versatility of the Covalent Thiol–Alkyne Reaction with Substituted Propargyl Warheads: A Deciding Role for the Cysteine Protease*. *J Am Chem Soc* **143**, 6423-6433 (2021); Ekkebus, R. et al. *On Terminal Alkynes That Can React with Active-Site Cysteine Nucleophiles in Proteases*. *J Am Chem Soc* **135**, 2867-2870 (2013); Sommer, S. et al. *Covalent inhibition of SUMO and ubiquitin-specific cysteine proteases by an in situ thiol–alkyne addition*. *Bioorg Med Chem* **21**, 2511-2517 (2013)). Upon insertion of the electrophilic moiety in C₃ instead of C₂, a significant decrease in probe reactivity was observed. However, we agree with the reviewer that an anti-Markovnikov addition is more common with the thiol-ene reaction and, considering we do not have a crystallographic structure showing thiol addition in C₂, the legend of Figure 1 has been modified from:

‘Figure 1. (...) b) Interaction between the HA-¹⁻⁷⁵Ub-alkene probe **1** and the DUB active site.’

To:

‘Figure 1. (...) b) Proposed interaction between the HA-¹⁻⁷⁵Ub-alkene probe **1** and the DUB active site.’

The Introduction has also been modified from:

‘An alkene moiety was installed in the place of the scissile C-terminal isopeptide bond, aligning the alkene moiety in the optimal position to react with the DUB active-site thiol upon its activation (Figure 1b)^{12,40,41}.’

To:

‘An alkene moiety was installed in the place of the scissile C-terminal isopeptide bond, aligning the warhead to react with the DUB active-site thiol in the C₂ position (Figure 1b)^{12,40,41,42}.’

The following reference was added:

42. Mons, E. et al. *Exploring the Versatility of the Covalent Thiol–Alkyne Reaction with Substituted Propargyl Warheads: A Deciding Role for the Cysteine Protease*. *J Am Chem Soc* **143**, 6423-6433 (2021)

2. Page 3, line 5. "The thiol-ene product 4 was obtained with DPAP and MAP, with an isolated yield of 63%". The yield is not the same with the one, 64%, in Table 1.

We appreciate this remark from the reviewer. The correct yield is 64% and the section was modified accordingly.

3. It would useful to list the chemical structures of Irgacure 2959 and Mes-Acr+.

We agree that this would be useful for the reader, and this opinion was also shared by reviewer 1. Figure 2 was added in response to this suggestion, please see the response to reviewer 1 for full details.